# Predictive value of increased C-reactive protein levels in preterm infants on respiratory function at five to six years of age

Mats Ingmar Fortmann [1,6], Rebecca Dappen[1,6], Claudia Roll[2], Margarita Kozhuharova[2], Axel von der Wense[3], Christoph Härtel[4], Julia Sandkötter[5], Egbert Herting[1], Wolfgang Göpel[1] & Alexander Humberg [5] ✉

## Abstract

**Background** Chronic lung disease in very low birthweight infants involves complex, inflammation-driven lung injury. This study aimed to assess whether C-reactive protein levels during the first 28 days of life predict lung function at five to six years of age.

**Methods** In this multicentre observational study, infants with a birthweight below 1500 grams and born between 2009 and 2015 were included. A C-reactive protein concentration above 10 mg/L was considered elevated. Recurrent elevations were defined as at least two values above this threshold separated by 14 days or more, with an interim decrease below 5 mg/L. Lung function was assessed by spirometry and running endurance at school age. Analyses included univariate tests and linear regression models adjusted for relevant risk factors.

**Results** Here we show that among 268 infants with a median gestational age of 27.6 (25.7–29.4) weeks, those with recurrent C-reactive protein elevations had higher rates of bronchopulmonary dysplasia (56.3% versus 29.4%, p < 0.027) and a greater proportion of children with forced expiratory volume in one second below the fifth percentile (71.9% versus 61.8%, p < 0.001), and reduced running endurance. Recurrent elevations showed a high positive predictive value (71.9%) but low sensitivity (20.2%) for detecting severely reduced lung function.

**Conclusions** The absence of recurrent C-reactive protein elevations in the neonatal period is associated with better long-term pulmonary outcomes and physical performance. These findings suggest that early inflammatory activity contributes to adverse lung development and support incorporating inflammatory biomarkers into multivariable models for early risk stratification.

## Plain language summary

Babies born very early and with very low birthweight often develop lung problems later in life. We studied whether a common blood marker of inflammation, called C-reactive protein, can help predict lung health and running endurance at school age. We followed infants born in several hospitals and measured their lung function and running endurance. We found that children who had repeated increases in C-reactive protein shortly after birth were more likely to have weaker lung function and lower physical endurance later on, while single increases did not have this effect. Our findings suggest that repeated inflammation early in life may have lasting consequences and highlight the importance of preventing or closely monitoring such episodes.

Premature birth affects a variety of organ systems with increased risk for medical morbidity and long-term sequelae[1]. One of the main complications is bronchopulmonary dysplasia (BPD), a serious respiratory complication affecting infants with disrupted alveolarization, microvascular development, thickening of the basement membrane and lymphocytic infiltration[2,3]. BPD is a multifactorial condition with diverse endotypes that include infection-inflammation-driven and placental dysfunction-related mechanisms. These endotypes contribute to distinct clinical phenotypes,

[1]Department of Paediatrics, University of Lübeck, Lübeck, Germany. [2]Department of Neonatology, Vest Children's Hospital Datteln, University Witten-Herdecke, Datteln, Germany. [3]Department of Paediatric Intensive Care and Neonatology, Altona Children's Hospital, Hamburg, Germany. [4]Department of Paediatrics, University of Würzburg, Würzburg, Germany. [5]Department of General Paediatrics, University Hospital Münster, Münster, Germany. [6]These authors contributed equally: Mats Ingmar Fortmann, Rebecca Dappen. ✉e-mail: alexander.humberg@ukmuenster.de

necessitating a shift toward precision medicine in BPD management. Emerging evidence highlights the potential of targeted therapeutic approaches tailored to these endotypes, as for example IL-1 receptor antagonists in mitigating inflammation-driven lung injury or anti-fibrotic agents like nintedanib for fibrotic endotypes of chronic lung disease[4,5].

The C-reactive protein (CrP), an acute phase protein, is the clinically most used inflammation and infection marker in neonates that rises several hours after the onset of inflammation[6]. Few studies described an association of postnatal elevated CrP levels with BPD[7,8]. From these observations it seems that an early detected CrP increase within the first days of life is associated with the development of BPD and can be taken as a biomarker for the prediction of BPD independently from the presence of either early- or late-onset sepsis[9,10]. Although BPD is a commonly used marker of lung injury, there is a growing consensus that it is a relatively poor predictor of long-term lung health. Children born preterm present with impaired lung function, regardless of a BPD diagnosis[11–13] and it is currently perceived that BPD is a relatively poor indicative factor for neonatal lung injury[14–16]. Given the limitations of BPD as a diagnostic marker, there is an urgent need for new, reliable predictive parameters that can identify VLBWI at high risk for chronic respiratory issues early on. Personalized management of BPD has the potential to improve long-term pulmonary outcomes by aligning therapeutic strategies with the specific molecular and clinical profiles of affected infants.

The aim of this study is to determine whether elevated C-reactive protein levels during the first 28 days of life, particularly when recurrent, are associated with impaired lung function in very low birthweight infants at five to 6 years of age. We show that repeated episodes of inflammation in the neonatal period are linked to poorer long-term pulmonary outcomes and reduced physical endurance, while single short-term elevations are not. These findings indicate that recurrent systemic inflammation contributes to adverse lung development and suggest that inflammatory biomarkers may support early risk stratification and personalized follow-up strategies in this vulnerable population.

## Methods
### Study population
The German Neonatal Network (GNN) is a multicentre observational population-based cohort study enrolling VLBWI with <1500 g birth weight from 2009–2016 and <1000 g birth weight since 2017 in Germany (www.vlbw.de)[17]. Yearly on-site-monitoring by a study nurse or paediatrician experienced in neonatology ensures proper assessment of clinical data. In the context of this study, VLBWI < 1500 g birth weight and extremely low birth weight infants (ELBWI, <1000 g birth weight) born in 4 GNN sites (University of Lübeck, Vest Children's Hospital Datteln, Altona Children's Hospital Hamburg, University Hospital Münster) between 1st of January 2009 and 31st of December 2015 and with information about laboratory CrP levels were included.

### Exclusion criteria included lethal abnormalities and missing data on CrP levels
**Ethics**. Approval by the local ethics committee for research in human subjects of the University of Lübeck (file number 08-022 and 14-220) and by the local ethics committees of all participating centres has been granted, specifically the Ethical Board of the Medical Chamber of the North Rhine region (2009021), Ethical Board of the Ruhr-University Bochum (3316-08), Ethical Board of the University of Aachen (EK020/11), Ethical Board of the University of Bonn (247/13), Ethical Board of the University of Erlangen-Nürnberg (157_15 Bc), Ethical Board of the Medical Chamber of the federal state of Mecklenburg-Vorpommern (A 2011 34), Ethical Board of the Medical Chamber of Bremen (RE/HR-398), Ethical Board of the Medical Chamber of Berlin (no separate file number, takeover by the ethics committee of the University of Lübeck), Ethical Board of the Medical Faculty Carl Gustav Carus Dresden (EK245082009), Ethical Board of the University of Düsseldorf (3122), Ethical Board of the University of Magdeburg (114/08), Ethical Board of

the University of Göttingen (16/12/11), Ethical Board of the University of Freiburg (451/11), Ethical Board of the University of Halle (2011-34), Ethical Board of the University of Tübingen (188/2011BO2), Ethical Board of the Medical School Hannover (365), Ethical Board of the University of Cologne (08-208), Ethical Board of the University of Jena (3592-10/12), Ethical Board of the University of Witten/Herdecke (52/2009), Ethical Board of the University of Essen (08-3926), Ethical Board of the Medical Chamber of the Westphalia-Lippe region (2008-469-b-S), Ethical Board of the Medical Chamber of Hamburg (MC-341/08), Ethical Board of the Medical Chamber of the federal state of Hessen (MC 253/2008), Ethical Board of the Medical Chamber of the federal state of Baden-Württemberg (B-2009-105-f), Ethical Board of the Medical Chamber of the federal state of Bavaria (7/08351), and Ethical Board of Saar University (188/08). The GNN was funded by the German Ministry for Education and Research (BMBF-grant-No: 01ER0805 and 01ER1501).

### Laboratory measurements of CrP values
CrP levels were measured using serum or plasma samples, carried out as directed by physicians by medical indication (e.g. suspicion of infection or monitoring the progress of a confirmed infection), and recorded retrospectively. The analysis of the CrP level was conducted using the standard Tina Quant CrP test and the Cobas c 701 analyzer from Roche. All participating centres applied identical protocols and internal quality controls to minimize inter-sample variability. The determination of the CrP concentration is accomplished through an immunological turbidimetry test.

CrP values were retrospectively imported into the GNN database from participating centres that provided laboratory access. Measurements were performed according to clinical indications rather than by a predefined sampling protocol.

### Definitions
CrP values were retrospectively transferred from local hospital laboratory systems into the GNN database. Recurrent CrP elevations were defined by at least two CrP values > 10 mg/L separated by a minimum interval of 14 days, with an interim decline below 5 mg/L to ensure that new peaks were distinguished from persistently elevated levels.

Bronchopulmonary dysplasia (BPD) was defined by the use of oxygen or respiratory support at 36 weeks postmenstrual age[18].

Of particular relevance to this study were the forced expiratory volume in one second (FEV1), the forced expiratory volume in one second as a percentage of predicted (FEV1%), and the forced vital capacity as a percentage of predicted (FVC%). Small for gestational age (SGA) was defined as birth weight <10th percentile according to population-based birth weight reference values[19]. Clinical chorioamnionitis was defined as suspected chorioamnionitis based on clinical signs as maternal fever, uterine tenderness, malodorous discharge, and maternal and fetal tachycardia with laboratory abnormalities (ie, leucocytosis).

### Follow-up
At the age of five to six years, participating children born preterm and included in the GNN were recruited for a structured follow-up evaluation. During the recruitment process, the study team reached out to the hospital where the children were born to schedule follow-up assessments. Families were randomly contacted and invited for follow-up with a particular emphasis on infants born before 28 weeks of gestational age. The follow-up examination encompassed various components, including interviewing the child's parents or caregivers, measuring the child's body parameters, neuromotor developmental assessment and lung function testing[20]. As part of the standardized GNN follow-up examination at 5 years of age, endurance performance was assessed using the 6-min run test. Each child was instructed to run or walk as far as possible within 6 min on a marked flat indoor or outdoor track under standardized supervision. The total distance covered (in meters) was recorded. Results were converted to age- and sex-specific percentiles based on reference data from the German health and

fitness surveys (KiGGS). For analysis, distance running <5th percentile was defined as markedly reduced endurance performance, and <15th percentile as below-average physical fitness. However, because distance running was only added to the study methods later in the follow-up studies, only part of the data could be analyzed in this regard.

### Measurement of lung function at five to six years

Lung function parameters were assessed using spirometry involving the utilization of a flow sensor and the Easy-on PC software (manufactured by ndd Medizintechnik AG, Zurich, Switzerland). The execution of lung function measurements followed a standardized protocol, commencing with a 10-min resting phase. Subsequently, the child assumed a comfortable sitting position on a chair. Nasal flow was prevented by a nose clip. Recording the parameters requires maximal expiration, which was achieved through visual representations on a notebook. Children were encouraged to blow out candles on the notebook, inflate a balloon as fully as possible with a single breath, or set a swing in motion through exhalation. Valid attempts involved maximal expiration to zero flow, followed by deep inhalation and a strong exhalation. Additionally, the quality of attempts was noted based on the criteria described above, as well as cooperation during the procedure and any remarks regarding potential accompanying factors, issues, or anomalies. Up to ten attempts could be made by each child with the best attempt being considered for analysis. In a secondary analysis, all results were again thoroughly checked by the authors and only data of children with technically acceptable and reproducible results were included in the analysis. Children unable to perform valid tests despite repeated attempts were excluded. Measurements were considered technically acceptable if they showed a rapid start of exhalation, no artefacts such as coughing or premature termination, and a clear end-of-test plateau. Results were deemed reproducible when at least two acceptable and comparable manoeuvres were obtained.

### Statistics and Reproducibility

The baseline characteristics of maternal and neonatal variables were represented in the form of medians, interquartile ranges (IQR), counts, frequencies, and 95% confidence intervals (CI) for column percentages. Unadjusted comparisons were assessed using the Chi-square test and the Mann-Whitney U-Test.

$FEV_1$ and FVC were documented in litres. The $FEV_1$ and FVC z-scores were calculated according to the Global Lung Function Initiative[21] with z-scores $< -1.644$ according to values <5th percentile. Linear regression models were computed for the z-scores of $FEV_1$ and FVC and adjusted for gestational age, birth weight, born small-for-gestational age, antenatal steroid usage, cerebral hemorrhage, periventricular leucomalacia, cerebral palsy (Gross Motor Function Classification System ≥1), surgical treatment of necrotizing enterocolitis, mechanical ventilation, duration of mechanical ventilation, duration of oxygen therapy within the first 28 days, presence of BPD, intelligence quotient, use of postnatal steroids (dexamethasone and/or hydrocortisone) and weight at five year follow-up. The diagnostic performance of recurrent CrP elevations was assessed using sensitivity, specificity, positive predictive value (PPV), and negative predictive value (NPV) for the outcome $FEV_1$ z-score <5th percentile.

All statistical analyses were carried out using SPSS software (IBM SPSS Statistics for Windows, Version 29.0, Munich, Germany). Figures were created using python language version 3.13.0 with matplotlib v3.9.3[22]. Raw data were generated at the University of Lübeck and the University of Münster. Derived data supporting the findings of this study are available from the corresponding author on request.

## Results

### Study population

Between January 1st 2009 and December 31st 2015, 13,849 infants <1500 g birth weight were enrolled in the GNN (shown in Fig. 1). Of these infants, 3848 were assessed for follow-up at the age of five to six years. After exclusion of infants not evaluated for CrP levels as these centres did not

participate in CrP value collection, primary datasets of 353 preterm infants with five to six year follow-up and CrP measurement remained for analysis. Among the 353 available datasets, 268 children met the secondary quality control standards for lung function testing, demonstrating reproducible and technically acceptable results.

Baseline data of infants with single elevation of Crp did not show differences to those with no elevations of CrP. However, differences were more evident in children with multiple CrP elevations. These infants were born at lower gestational age (25.8 [24.5–27.4] weeks vs. 27.9 weeks [26.1–29.7], $p < 0.001$) and birth weight (723 g [560–905] vs. 980 g [770–1249], $p < 0.001$) with higher rates of born SGA (31.3% [17.3–48.4] vs. 12.4% [8.4-17.4], $p = 0.005$) and increased rates of neurologic and abdominal complications (supplementary table 1).

### Univariate analyses

VLBWI with single CrP elevation had significantly higher rates of oxygen need at discharge (5.9% [1.2–17.6] vs. 0 %, $p = 0.001$) and higher rates for z-scores <5th percentile for $FEV_1$ (61.8% [45.0–76.6] vs. 34.7% [28.3–41.4], $p < 0.001$) and FVC (50.0% [33.8–66.2] vs. 36.6% [30.2–43.4], $p < 0.001$) (see supplementary table 2).

In comparison to cases with no elevation of CrP, the group exhibiting recurrent elevation of CrP showed significantly higher incidences of BPD (56.3% [39.1–72.3] vs. 16.8% [12.0–22.4], $p < 0.001$), increased oxygen requirement at discharge (12.5% [4.4–27.0] vs. 0%, $p < 0.001$), higher rates of non-invasive ventilation at discharge (6.3% [1.3–18.6] vs. 0.5% [0.1–3.0], $p = 0.007$), a lower median weight at five to six-year follow-up (16.7 kg [15.1–19.2] vs. 18.6 kg [17.0–20.8], $p = 0.002$) and higher rates for gross motor function scale (GMFCS) scores ≥1 (34.5% [19.3–52.6] vs. 16.2% [11.4–22.0], $p = 0.019$). Regarding respiratory function and related parameters, recurrence of elevated CrP was associated with reduced lung function as evidenced by decreased $FEV_1$ (0.77 [0.67–0.91] vs. 0.98 litres [0.85–1.12], $p < 0.001$). Moreover, z-scores of $FEV_1$ and FVC were significantly reduced in infants with recurrent elevations and the percentage of individuals with $FEV_1$ and FVC z-score <5th percentile is markedly higher in the recurrent elevation of CrP group (71.9% [54.9–85.1] vs. 34.7% [28.3–41.4], $p < 0.001$) and (81.3% [65.4–91.8] vs. 36.6% [30.2–44.4], $p < 0.001$), respectively.

At the 5–6-year follow-up, children with recurrent CrP elevations showed markedly poorer endurance performance. 22.2% of these children performed below the 5th percentile in the 6-min run test, compared with 7.7% among those without CrP elevations ($p = 0.021$). However, as not all children could be followed up regarding distance running, only $n = 133$ datasets could be analyzed.

VLBWI with recurrent CrP elevations differed significantly from children with a single CrP elevation with respect to BPD rate (56.3% [39.1–72.3] vs. 29.4% [16.2–45.9], $p = 0.027$), and median weight at 5 years of age (16.7 kg [15.1–19.2] vs. 18.8 kg [16.7–19.4], $p = 0.027$), as well as reduced median lung parameters for FEV (0.77 l [0.67–0.91] vs. 0.93 [0.80–1.06], $p = 0.009$) and FVC (0.8 l [0.66–0.94] vs. 1.01 l [0.83–1.15], $p = 0.002$) and a higher rate of z-score values <5th percentile for FVC (81.3% [65.4–91.8] vs 50.0% [33.8-66.2], $p = 0.008$).

The prevalence of asthma or obstructive bronchitis during the 12 months before the 5–6-year follow-up did not differ significantly between groups, with similar rates among children with recurrent (31.0%) and single (36.4%) CrP elevations compared to those without elevations (30.6%; $p = 0.658$).

### Adjusted analyses

To adjust for possible confounding variables, z-scores of lung function were tested in a linear regression model. Scatterplots with the fitted regression line are examined to ensure model assumptions were met. The residuals appeared to be independent (Durbin-Watson).

In this analysis, several factors exhibited associations with reduced lung function. Recurrent elevations of CrP correlated with decreased $FEV_1$ and FVC z-scores (Table 1 and Table 2), indicating that this factor was

**Fig. 1 | In- and exclusion of infants from the GNN for current analysis.** VLBWI = very low birthweight infants, CrP = C-reactive protein

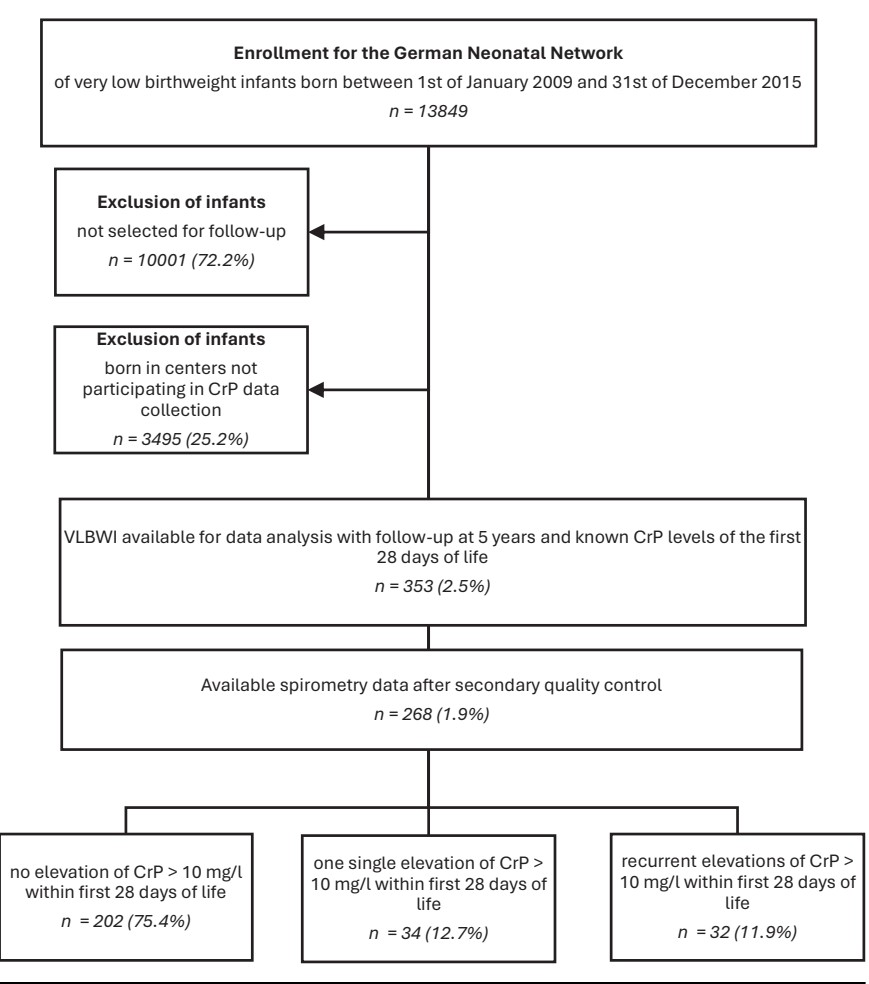

## Table 1 | Linear regression model for respiratory long-term outcome

| Variables | z-score of FEV$_1$ | | | |
|---|---|---|---|---|
| | *B* (SD) | Beta | *T* | *p* value |
| GMFCS ≥ 1 | −0.115 (0.197) | −0.041 | −0.6 | 0.561 |
| Weight at 5YFU [kg] | 0.012 (0.019) | 0.044 | 0.6 | 0.538 |
| IQ | 0.013 (0.006) | 0.148 | 2.2 | 0.028 |
| Duration of mechanical ventilation [days] | −0.213 (0.083) | −0.282 | −2.6 | 0.011 |
| Recurrent CrP > 10 mg/l | −0.617 (0.251) | −0.184 | −2.5 | 0.015 |

Model further adjusted for gestational age, birth weight, antenatal administration of steroids, IVH, surgical treatment for necrotizing enterocolitis, birth weight <10th percentile, duration of mechanical ventilation, duration of oxygen therapy within the first 28 days, use of postnatal corticosteroids (dexamethasone and/or hydrocortisone), BPD and PVL (data not shown). $R^2$: 0.270; Durbin-Watson: 1.737; $F$ = 4.480 with $p$ < 0.001; Abbreviation: B: unstandardized coefficients, Beta: standardized coefficients, IQ: intelligence quotient.

## Table 2 | Linear regression model for respiratory long-term outcome

| Variables | z-score of FVC | | | |
|---|---|---|---|---|
| | *B* (SD) | Beta | *T* | *p* value |
| GMFCS ≥ 1 | −0.524 (0.220) | −0.160 | −2.4 | 0.018 |
| Weight at 5YFU [kg] | 0.045 (0.021) | 0.146 | 2.1 | 0.035 |
| IQ | 0.013 (0.007) | 0.124 | 1.9 | 0.055 |
| Duration of mechanical ventilation [days] | −0.195 (0.093) | −0.224 | −2.1 | 0.036 |
| Recurrent CrP >10 mg/l | −0.991 (0.280) | −0.256 | −3.5 | 0.001 |

Model further adjusted for gestational age, birth weight, antenatal administration of steroids, IVH, surgical treatment for necrotizing enterocolitis, birth weight <10th percentile, duration of oxygen therapy within the first 28 days, use of postnatal corticosteroids (dexamethasone and/or hydrocortisone), BPD and PVL (data not shown). $R^2$: 0.317; Durbin-Watson: 1.939; $F$ = 5.624 with $p$ < 0.001; Abbreviation: B: unstandardized coefficients, Beta: standardized coefficients, IQ: intelligence quotient.

linked to decreased lung function. However, single increases of CrP were not correlated with reduced FEV$_1$ and FVC values (see Supplementary Table 3a, 3b).

Notably, duration of ventilation demonstrated a positive correlation with FEV$_1$ and FVC z-scores, suggesting that longer mechanical ventilation impacts spirometry values in later life. This result highlighted the importance of ventilation strategies, while also suggesting the negative impact of

recurrent elevations in CrP levels on the respiratory function of these individuals in their early childhood years.

### Accuracy of tests
Testing the predictive performance of recurrent CrP >10 mg/L within the first 28 days for the outcome FEV$_1$ z-score below the 5th percentile yielded a sensitivity of 20.2 % and a specificity of 94.2 %. The positive predictive value

(PPV) was 71.9 %, and the negative predictive value (NPV) was 61.4 %, indicating that recurrent CrP elevations were highly specific but not sensitive markers of severely reduced long-term lung function.

## Supplementary analyses

Frequency of CrP determinations grouped by recurrent elevations > 10 mg/l against all other infants with single and no CrP elevation is given in Supplementary Fig. 1. Highest frequencies of CrP determination are found within the first days of life. Infants with recurrent CrP elevations have continuously higher values of CrP during the first 28 days of life (supplementary fig. 2) and higher values of inflammatory markers in the white blood count (Supplementary Table 4), indicating a higher systemic inflammatory impact. Furthermore, infants with worse respiratory long-term outcome show higher median CrP levels over the first 28 days of life (Supplementary Fig. 3 and 4). However, median levels of CrP were not associated with worse respiratory outcome (Supplementary Table 5a, 5b). Independent from early diagnosis of BPD, frequencies for long-term respiratory sequalae are increased represented in the group of recurrent CrP elevations (supplementary table 6).

## Ethics approval and consent to participate

This study was conducted in accordance with the Declaration of Helsinki and approved by the Ethics Committee of the University of Lübeck (file numbers 08-022 and 14-220) and the ethics committees of all participating centres. Informed consent was obtained from the parents or legal guardians of all participants.

## Discussion

This study investigated the association between elevated CrP levels within the first 28 days of life in VLBWI and long-term respiratory function at the age of five to six years. Our findings revealed a significant correlation between recurrent elevation of CrP during the neonatal period and adverse respiratory outcomes in later life. We could further show that the absence of recurrent CrP elevations has potential at correctly excluding individuals without reduced lung function. However, the low sensitivity indicates a considerable proportion of true cases are missed and limits its utility as a standalone diagnostic tool for identifying positive cases. We here used the measurement of $FEV_1$ and FVC via spirometry as a standard pulmonary function test for neonatal outcome-measurement[23].

Furthermore, children at school age with recurrent CrP elevations demonstrated significantly reduced endurance performance, suggesting that early systemic inflammation may contribute not only to measurable lung function impairment but also to reduced overall physical fitness. This observation aligns with the hypothesis that neonatal inflammatory processes can interfere with normal pulmonary and cardiovascular development, leading to persistent functional limitations beyond airway obstruction alone. In combination with the spirometric findings, these results support the concept that recurrent neonatal inflammation represents an early-life determinant of long-term respiratory and exercise capacity in preterm survivors. In contrast, the prevalence of asthma or obstructive bronchitis did not differ between groups, indicating that reduced endurance is unlikely to be explained solely by current respiratory morbidity. However, information on early childhood respiratory infections or wheezing episodes was not available, which limits interpretation of potential intermediary mechanisms.

Our study aligns with previous research linking early inflammation or immune-dysregulation[24] to adverse respiratory outcomes. Stimulation of inflammatory cascades, involving cytokines and pattern recognition receptors, can lead to endothelial cell activation, oxidative stress, cell death, and microvascular complications, contributing to lung injury in neonates[25–28]. Studies have shown associations between elevated levels of serum eosinophil chemotactic factors, TNF-α, IL-1β, IL-6, IL-8, and other pro-inflammatory markers with the occurrence and prognosis of BPD[29]. Furthermore, reductions in neonatal sepsis prevalence appear to correlate with a decreased incidence of BPD, suggesting a potential link between postnatal sepsis and the risk of BPD in premature infants[30]. Rodent models

of postnatal sepsis-induced lung injury using systemic LPS exposure in newborn mice highlight the vulnerability and an ontogenic window in early lung development to inflammation-induced disruptions, affecting lung morphogenic pathways critical for distal acinar development[27,28,31]. These data underscore the role of endothelial cells in both developmental angiogenesis and sepsis-induced dysmorphic angiogenesis, indicating that factors like vascular endothelial growth factor (VEGF), angiopoietins, and FOSL1 play complex roles in this process[32–34]. An altered elastic fibre assembly, mesenchymal and fibroblast growth factor downregulation, and protease-mediated lung extracellular matrix degradation also play an important role in sepsis-induced lung injury[35,36].

Induction of CrP expression is significantly influenced by IL-6 and IL-1, leading to activation of phagocytic cells, production of inflammatory cytokines, and regulation of the complement pathway, potentially resulting in adverse vascular events when elevated[37]. In clinical settings, monitoring of CrP levels is one of the most widely used detection methods for neonatal sepsis. The relationship between CrP and diseases, particularly in premature infants, has garnered significant attention as it may aid to early detect a risk for respiratory long-term complication and may guide future interventions for BPD to improve prognoses. Studies have consistently linked elevated CrP levels with bronchopulmonary dysplasia (BPD) in preterm infants[7]. Elevated CrP levels, particularly on day 28 of life, have been associated with BPD and increased mortality risk, indicating a systemic inflammatory response[7]. Notably, elevated CrP levels have been observed early before clinical symptoms of BPD manifestation, highlighting its potential as an early diagnostic marker[10].

One strength of this study is that results of respiratory outcome are adjusted for several risk factors impacting respiratory outcome, like gestational age, birth weight and mechanical ventilation. Besides these variables, increased recurrent CrP values are associated with poor expiratory airflow at age five to six years. Furthermore, evaluation of lung function parameters assessed by a consistent team of physicians and nurses suggests a high consistency of the data. Despite its strengths, our study might have limitations worth considering. For instance, the focus on expiratory airflow as an outcome might not encompass the entirety of respiratory function. Additionally, the absence of detailed mechanistic insights or direct causality between long-term lung impairment and increased CrP values might warrant further investigations. Furthermore, studies have indicated a systemic inflammatory response associated with Ureaplasma spp. in premature neonates, leading to elevated CrP levels. This chronic systemic inflammatory response may be linked to the development and persistence of lung injury. We here also cannot control for potential effects as the presence for ureaplasma spp. is not recorded in our dataset. The retrospective nature of our study may introduce bias and limit the establishment of causal relationships. The absence of planned blood samples in the GNN dataset constrained the availability of CrP data, potentially affecting the representativeness of the sample. Additionally, the study's reliance on CrP as a sole inflammatory marker may overlook the contribution of other inflammatory factors. Because the type of sample (serum or plasma) was not uniformly documented, minor systematic differences between sample matrices cannot be excluded. CrP determinations were obtained based on clinical indications and retrospectively incorporated from several centres, which may have led to overrepresentation of infants with higher illness burden. Nevertheless, the observed association between recurrent CrP elevations and reduced lung function remained significant after adjustment for major confounders, supporting the robustness of our findings. Our follow-up cohort has a risk of selection bias. For the follow-up, we chose a random invitation practice. However, it is possible that the invitation process and voluntary participation in the follow-up studies resulted in a bias towards presenting an incompletely balanced group. Only a small proportion of the overall GNN cohort was included in this analysis. Consequently, infants in this analysis may differ socioeconomically and clinically from those not included, which could limit generalizability. Nevertheless, the observed associations between recurrent CrP elevations and reduced lung

function likely remain valid within the analysed subgroup, although replication in larger, prospectively collected cohorts is warranted. Furthermore, while efforts are made to adjust for confounding variables, residual confounders might influence the observed associations. Further studies should therefore take additional factors as environmental exposures like genetic predispositions, histologic proved chorioamnionitis, hemodynamic relevant factors during prematurity, respiratory infections, parental smoking or vaccination status into account[38]. Because spirometry performance in preschool-aged children depends on effort and neurodevelopmental ability, variability in cooperation remains a limitation. Although only technically acceptable and reproducible results were analysed, children with developmental impairments may have been underrepresented in the final analysis. Lung function was assessed at a single standardized time point (five to six years of age). Without earlier or later measurements, it remains uncertain whether the observed impairments represent permanent airway dysfunction or a temporary alteration in lung growth trajectories. Future longitudinal follow-up within the GNN cohort will be crucial to clarify the evolution of pulmonary function into later childhood and adolescence. Beyond $FEV_1$ and FVC, parameters such as $FEF_{25-75}$% may more sensitively reflect small airway involvement in children born preterm. However, this parameter was not systematically recorded in our cohort, and the available data were insufficient for analysis. Because only surviving infants were eligible for follow-up, the sickest neonates with the highest inflammatory burden were not represented in the present analysis. This survivor bias may have led to an underestimation of the true strength of associations between early systemic inflammation and long-term pulmonary dysfunction. Nonetheless, significant associations observed within the surviving cohort underscore the relevance of inflammatory processes even among relatively stable preterm infants.

From our data it can be assumed, that infections detected by increased CrP levels increase the risk for developing respiratory complications. From our data we cannot distinguish between recurrent infections reflected by different CrP elevations or a possible underlying sustained inflammation leading to a continuous process of inflammatory process. Although dichotomizing CrP levels at >10 mg/L simplifies interpretation, it may mask the contribution of inflammation magnitude and duration. In supplementary analyses using continuous and trajectory-based representations, higher median CrP values were observed among children with impaired lung function, but statistical significance was not retained after adjustment. These findings furthermore suggest that recurrent elevations, reflecting repeated inflammatory episodes, could be more predictive of adverse pulmonary outcomes than isolated high values. Although our definition of recurrent CrP elevation required a temporary normalization followed by renewed increase, we cannot fully distinguish whether this pattern represents repeated inflammatory episodes or fluctuating low-grade inflammation. This limitation highlights the need for future studies with prospectively standardized sampling intervals to capture the dynamics and mechanisms of early-life inflammation in preterm infants more precisely.

In conclusion, our study conveys information about the impact of systemic inflammation on later respiratory outcomes in preterm infants. Our findings highlight the importance of early inflammatory markers in predicting long-term respiratory health and warrant further research to explore interventions for mitigating the impact of inflammation on neonatal lung injury and subsequent respiratory complications. Our findings emphasize the need for individualized approaches to monitoring neonatal inflammation and its progression. This perspective aligns with the growing interest in tailoring treatment strategies based on endotype classification[39]. Interventions targeting the inflammatory pathways, such as IL-1 receptor antagonists like anakinra, hold promise in mitigating inflammation-driven lung injury[40,41]. While BPD has traditionally been used as a predictor, it is now recognized as a relatively poor measure of neonatal lung injury[42]. Therefore, there is a need for new and feasible diagnostic methods to better understand lung injury in these vulnerable infants and to predict their future respiratory health accurately. This knowledge is essential for assessing interventions aimed at early prevention and treatment of lung disease in preterm infants. The measurement of recurrent CrP elevations in a combined model with other risk factors such as mechanical ventilation could be a clinical tool for further risk assessment studies. Future prospective studies are needed to validate its prognostic utility. Further research is warranted to elucidate the mechanisms linking early inflammation to CLD development and to evaluate the feasibility of integrating inflammatory monitoring into clinical practice. Prematurity itself represents the primary determinant of altered lung growth and structure, predisposing to long-term respiratory morbidity. In this context, recurrent postnatal inflammatory episodes—as indicated by elevated CrP levels—should be regarded as modifiers that exacerbate an already vulnerable developmental trajectory rather than as independent causative agents. Understanding the interaction between baseline immaturity and secondary inflammatory stressors will be essential for developing individualized prevention and treatment strategies. Although CRP alone is not a sufficient predictive biomarker, our findings underscore the potential clinical relevance of preventing repeated inflammatory stress in early postnatal life.

From our data it can be assumed that infections detected by increased C-reactive protein levels increase the risk for developing respiratory complications. However, it remains uncertain whether these elevations reflect distinct recurrent infections or an underlying persistent inflammatory process. Although dichotomizing C-reactive protein levels at 10 milligrams per litre simplifies interpretation, it may obscure the impact of inflammation magnitude and duration. Supplementary analyses using continuous representations suggested that higher median C-reactive protein values are associated with impaired lung function, although statistical significance was not maintained after adjustment. These findings support the concept that recurrent elevations, rather than isolated peaks, are more predictive of adverse pulmonary outcomes. Future studies with prospectively standardized sampling intervals are needed to better characterize the dynamics of early-life inflammation in preterm infants. In summary, this multicentre study demonstrates that recurrent elevations of C-reactive protein during the neonatal period are associated with poorer lung function and reduced running endurance at school age in children born very preterm. These findings indicate that repeated systemic inflammation in early life contributes to impaired pulmonary and physical development, whereas single short-term inflammatory episodes do not appear to have lasting effects. The absence of recurrent inflammation is linked to more favorable long-term outcomes, emphasizing the potential of inflammatory biomarkers for identifying infants at risk. Incorporating C-reactive protein patterns into multivariable risk models may enhance early risk stratification and guide individualized follow-up and preventive strategies. Future prospective studies should validate these findings and explore interventions aimed at limiting recurrent inflammatory stress in this vulnerable population.

## Data availability

Raw data of the German Neonatal Network were generated at the University of Luebeck. The datasets generated and analyzed during the current study are available from the corresponding author upon reasonable request. Data are stored as SPSS and CSV files on secure institutional servers at the University Hospitals Lübeck and Münster. Access to these data is controlled due to patient privacy and ethical restrictions. Requests for access should be directed to PD Dr. med. Alexander Humberg (alexander.humberg@ukmuenster.de), University Hospital Münster, Department of General Pediatrics, Neonatology & Pediatric Intensive Care, Albert-Schweitzer-Campus A1, D-48149 Münster, Germany. Requests will be reviewed within four weeks, and data sharing will be subject to institutional data use agreements permitting non-commercial scientific use only.

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

## Acknowledgements

We sincerely thank all participating children and their families for their invaluable contribution to this study. We also acknowledge the dedicated

medical, nursing, and research staff of the German Neonatal Network for their ongoing commitment to data collection and follow-up. The German Neonatal Network was funded by the German Ministry for Education and Research (Bundesministerium für Bildung und Forschung, BMBF; grant numbers 01ER0805 and 01ER1501). The funder had no role in study design, data collection, data analysis, interpretation, or the decision to submit the manuscript for publication.

## Author contributions

M.F., R.D., and A.H. wrote the manuscript; R.D., W.G., and A.H. performed the analytic calculations; A.H., C.H. E.H., and W.G. planned the study; M.F., R.D., C.R., M.K., A.W., C.H., J.S., and A.H. performed the data acquisition, M.F., R.D., C.R., A.W., C.H., E.H., W.G., and A.H. contributed to the design and implementation of the research. All authors discussed the results and commented on the manuscript.

## Funding

## Competing interests

The authors declare no competing interests.
