## [Transparent Peer Review file · Communications Medicine]

Predictive value of increased C-reactive protein levels in preterm infants on respiratory function at five to six years of age

Corresponding Author: Dr Alexander Humberg

Version 0:

Reviewer comments:

Reviewer #1

(Remarks to the Author)

The predictive significance of elevated C-reactive protein levels in preterm infants regarding respiratory function at five to six years of age: a cohort study.

Summary

This multicenter cohort study, utilising data from the German Neonatal Network, assesses whether recurrent elevations of C-reactive protein (CrP) in very low birthweight infants (VLBWI) during the initial 28 days of life can predict long-term pulmonary function at 5–6 years of age. In a cohort of 353 infants with neonatal C-reactive protein (CrP) data and subsequent spirometry, recurrent CrP elevations (>10 mg/L, with intervals of ≥ 14 days) correlated with increased incidence of bronchopulmonary dysplasia (BPD), heightened requirements for ventilation and oxygen therapy, and diminished lung function, as indicated by lower FEV1 and FVC z-scores. The lack of recurrent CrP elevations exhibited a significant negative predictive value for impaired lung function, whereas the positive predictive value was minimal. The study indicates that systemic inflammation could significantly influence long-term respiratory outcomes in preterm infants and underscores the potential function of CrP in early risk stratification.

Concerns:

1. The research utilised both serum and plasma samples for CrP analysis. Because plasma values are usually higher, using a standard cut-off of 10 mg/L could cause errors in classification. Sensitivity analyses that separate serum and plasma values or take into account the type of sample would make the results more reliable.
2. Dichotomising CRP (>10 mg/L) loses information about magnitude and duration. Continuous or trajectory-based approaches could provide richer insights.
3. CrP measurements were taken based on clinical indications instead of systematically, which could skew the dataset towards infants with a greater illness burden.
4. It is still not clear if repeated CrP spikes are caused by repeated infections or long-lasting inflammation throughout the body. This distinction is significant for informing future interventions but remains unaddressed.
5. This analysis only included 2.5% of the original group. Infants with complete data may systematically differ from those excluded, raising concerns regarding representativeness. Families returning for follow-up often differ socioeconomically and clinically, potentially biasing results.
6. BPD was based on the need for oxygen at 36 weeks postmenstrual age. Nonetheless, the desired oxygen saturation level was not delineated. Because different centres may use different thresholds, it's not clear how comparable the BPD diagnosis is across sites.
7. In this age group, spirometry depends on how hard the person tries. The manuscript discusses motivational tools and various attempts; however, the reproducibility of FEV1 and FVC measurements continues to be a concern, especially for children with neurodevelopmental impairments. How many kids passed technically acceptable tests?
8. When following up with children who have BPD, FEF_{25–75%} is a better sign of small airway dysfunction than FEV1 by itself. Although FEV1 and FEV1/FVC are fundamental to spirometry interpretation, they may not accurately reflect subtle small airway impairment. Moreover, physical fitness markedly affects spirometric performance in survivors of BPD. Numerous studies (e.g., Vrijlandt et al., Thorax 2006; Welsh et al., Eur Respir J 2010; Clemm et al., Acta Paediatr 2021; Praprotnik et al., Pediatr Pulmonol 2023) have demonstrated that diminished exercise tolerance and cardiorespiratory fitness are associated with decreased spirometric values. This indicates that the observed decline in lung function may be

attributable to deconditioning rather than inherent pulmonary pathology. The manuscript ought to address this factor in the analysis of the long-term spirometry data.

9. Only one follow-up point (5–6 years) was available. Without earlier or later measures, it is unclear whether deficits reflect a permanent impairment or altered growth trajectories.

10. There is no information on how many obstructive episodes (e.g., viral wheezing, asthma exacerbations, hospitalisations for lower respiratory tract illness,...) the children experienced between discharge and follow-up at 5–6 years. These episodes are known to influence spirometric outcomes (FEV₁, FEV₁/FVC) in preterm populations (Kotecha et al., Thorax 2013; Simpson et al., Eur Respir J 2015). Their omission limits interpretation, as impaired lung function may reflect cumulative morbidity rather than only neonatal inflammation.

11. There is no data on postnatal corticosteroid use, which is a significant factor affecting the incidence of BPD and may vary between centres. This omission restricts interpretation, as steroid exposure may obscure associations between CrP, BPD, and long-term pulmonary outcomes.

12. Prematurity itself should be acknowledged as an underlying driver of abnormal lung development, with CRP elevations and other neonatal exposures acting as modifiers of an already disrupted process.

13. The sickest infants with the highest inflammatory burden may not have survived to follow-up, potentially underestimating the strength of associations.

14. Regression and ROC analyses are suitable; however, various concerns diminish confidence:

- Small subgroup sizes (only 42 with recurrent CrP elevations) may produce unreliable estimates.
 - A number of univariate and adjusted tests were done, but no adjustments were made for multiplicity.
 - Data that were missing were "ignored," but there are no details about how much or why they were missing.
 - Collinearity among related variables, such as ventilation duration, oxygen therapy, and BPD, was not considered.
 - ROC analysis demonstrated limited predictive capability, yielding a very low positive predictive value (PPV) of 19.8%.
- Combined predictive models (e.g., CrP + ventilation days + BPD) may more accurately represent clinical application.

15. This analysis only included 2.5% of the original group. Infants with complete data may systematically differ from those excluded, raising concerns regarding representativeness. Families returning for follow-up often differ socioeconomically and clinically, potentially biasing results.

Strengths

- The study features a substantial multicenter cohort and standardised follow-up evaluations.
- The study accounts for various pertinent confounders, including gestational age, birth weight, ventilation, BPD, and IQ.
- The study prioritises long-term respiratory function over merely diagnosing BPD.

Suggestion

This research offers significant evidence connecting systemic inflammation to negative pulmonary outcomes in preterm infants. The work is biologically plausible and consistent with prior literature, and the long-term focus on spirometry at 5–6 years is a strength.

Nonetheless, several aspects require further clarification or discussion: measurement techniques (serum vs. plasma CRP), the definition of BPD (oxygen saturation targets not specified), absence of data on postnatal steroid exposure, reliance on FEV₁/FVC without small airway indices, statistical limitations (small subgroups, missing data, no adjustment for collinearity or multiplicity), and the narrow scope of outcome assessment (spirometry only, no symptoms or obstructive episodes).

While not all of these issues can be corrected with the current dataset, they should be acknowledged and discussed more explicitly. Doing so will strengthen the manuscript by improving transparency, contextualising the findings, and guiding future research directions. Your findings highlight an important practical point: the absence of recurrent CrP elevations appears protective, reducing the risk of long-term lung impairment in preterm infants. It would be beneficial to make this more evident in the discussion, as it strengthens the clinical relevance of your work, even if CRP alone is not sufficient as a predictive biomarker.

Reviewer #2

(Remarks to the Author)

As a multicenter retrospective study in VLBWI, this research features a long observation period and relatively complete data, providing meaningful support for clinical decision-making. Moreover, it suggests that CRP may have potential utility in the risk stratification of chronic lung disease, although its accuracy requires further validation. Overall, the authors and their team have made a valuable contribution, and the study is suitable for publication.

Version 1:

Reviewer comments:

Reviewer #1

(Remarks to the Author)

The authors have satisfactorily addressed the majority of the reviewers' suggestions and have provided appropriate clarifications regarding the identified shortcomings and potential sources of bias. The revisions have improved the clarity and power of the manuscript. At this stage, I have no further substantive comments.

Reviewer Comments to COMSMED-25-1883A:

Predictive value of increased C-reactive protein levels in preterm infants on respiratory function at five to six years of age

Running title: C-reactive protein and lung function in preterm infants

Mats Ingmar Fortmann^{1,*}, Rebecca Dappen^{1,*}, Claudia Roll², Margarita Kozhuharova², Axel von der Wense³, Christoph Härtel⁴, Julia Sandkötter⁵, Egbert Herting¹, Wolfgang Göpel¹ and Alexander Humberg⁵

¹ Department of Paediatrics, University of Lübeck, Lübeck, Germany

² Department of Neonatology, Vest Children's Hospital Datteln, University Witten-Herdecke, Datteln, Germany.

³ Department of Paediatric Intensive Care and Neonatology, Altona Children's Hospital, Hamburg, Germany.

⁴ Department of Paediatrics, University of Würzburg, Würzburg, Germany.

⁵ Department of General Paediatrics, University Hospital Münster, Münster, Germany

* These authors contributed equally to this work

Reviewer#1

Comment 1: The research utilised both serum and plasma samples for CrP analysis. Because plasma values are usually higher, using a standard cut-off of 10 mg/L could cause errors in classification. Sensitivity analyses that separate serum and plasma values or take into account the type of sample would make the results more reliable.

We thank the reviewer for this important comment. We agree that C-reactive protein (CrP) concentrations can differ slightly depending on whether serum or plasma samples are used, as plasma values tend to be marginally higher due to fibrinogen interference during measurement. In our study, CrP determinations were performed using the Tina-Quant CRP assay on the Roche Cobas c701 analyzer, which is validated for both serum and plasma with a standardized calibration curve and comparable analytical performance between sample types.

Unfortunately, we did not record whether CrP was measured from serum or plasma for each individual measurement. However, all participating laboratories used identical assay platforms and standardized internal quality controls according to the manufacturer's specifications, minimizing inter-site and inter-sample variability.

To address this concern, we have now added a statement to the Methods section clarifying the analytical process. We have also added a sensitivity note to the Limitations section acknowledging that minor differences between serum and plasma CrP levels cannot be fully excluded and may have contributed to minimal measurement variability, though this is unlikely to have substantially influenced the overall classification or the direction of associations observed. We furthermore added an analysis (linear regression) to this review by calculating the role of the different clinic on CrP values. 2 of 4 hospitals used only one method of CrP analysis in preterm infants (serum or plasma samples). The other 2 hospitals showed mixed set ups. In the linear regression the role of the participating NICU has no influence on CrP value. Therefore, we did not add this calculation to the manuscript.

Table: Linear regression model for respiratory long-term outcome in dependence of the clinic of birth (table not in manuscript)

Variables	z-score of FEV ₁			
	B (SD)	Beta	T	p-value
Clinic of birth	-0.008 (0.006)	-0.099	-1.395	0.165
Recurrent CrP > 10 mg/l	-0.585 (0.252)	-0.174	-2.323	0.021

Table: Linear regression model for respiratory long-term outcome in dependence of the clinic of birth. Model further adjusted for GMFCS ≥ 1 , Weight at 5YFU [kg], IQ, Duration of mechanical ventilation [days], gestational age, birth weight, antenatal administration of steroids, IVH, surgical treatment for necrotizing enterocolitis, birth weight < 10th percentile, duration of mechanical ventilation, duration of oxygen therapy within the first 28 days, BPD and PVL (data not shown). R^2 : 0.277; $F=4.434$ with $p < 0.001$; Abbreviation: B: unstandardized coefficients, Beta: standardized coefficients, IQ: intelligence quotient

Table: Linear regression model for respiratory long-term outcome in dependence of the clinic of birth (table not in manuscript)

Variables	z-score of FVC			
	B (SD)	Beta	T	p-value
Clinic of birth	-0.011 (0.007)	-0.115	-1.677	0.095
Recurrent CrP > 10 mg/l	-0.947 (0.280)	-0.245	-3.383	< 0.001

Table: Linear regression model for respiratory long-term outcome in dependence of the clinic of birth. Model further adjusted for GMFCS ≥ 1 , Weight at 5YFU [kg], IQ, Duration of mechanical ventilation [days], gestational age, birth weight, antenatal administration of steroids, IVH, surgical treatment for necrotizing enterocolitis, birth weight < 10th percentile, duration of oxygen therapy within the first 28 days, BPD and PVL (data not shown). R^2 : 0.327; $F=5.501$ with $p < 0.001$; Abbreviation: B: unstandardized coefficients, Beta: standardized coefficients, IQ: intelligence quotient

We have added the following passages to our manuscript:

Methods, "Laboratory measurements of CrP values" (page 10, lines 164-165)
All participating centres applied identical protocols and internal quality controls to minimize inter-sample variability.

Discussion, "Limitations" (page 24, lines 418-419)

Because the type of sample (serum or plasma) was not uniformly documented, minor systematic differences between sample matrices cannot be excluded.

Comment 2: Dichotomising CRP (>10 mg/L) loses information about magnitude and duration. Continuous or trajectory-based approaches could provide richer insights.

We appreciate this insightful suggestion. We agree that treating C-reactive protein (CrP) as a dichotomous variable may reduce information about both the magnitude and temporal dynamics of inflammation. In our initial analyses, we chose the

established clinical cut-off of >10 mg/L to facilitate translational interpretation and comparability with previous neonatal studies linking elevated CrP to adverse outcomes (Ehl S, Gering B, Bartmann P, Högel J, Pohlandt F. C-reactive protein is a useful marker for guiding duration of antibiotic therapy in suspected neonatal bacterial infection. *Pediatrics*. 1997 Feb;99(2):216-21. doi: 10.1542/peds.99.2.216. PMID: 9024449.; Benitz WE, Han MY, Madan A, Ramachandra P. Serial serum C-reactive protein levels in the diagnosis of neonatal infection. *Pediatrics*. 1998 Oct;102(4):E41. doi: 10.1542/peds.102.4.e41. PMID: 9755278.).

To further address this point, we have conducted additional exploratory analyses considering median CrP values over the first 28 days of life as a continuous variable and comparing infants according to CrP trajectories (no, single, or recurrent elevations). These supplementary analyses are presented in Supplementary Table 4a–b and Figures 3–4, showing that while higher median CrP levels tended to be observed in infants with poorer respiratory outcomes, the association did not reach statistical significance after adjustment for confounders. This finding suggests that recurrent inflammatory episodes, rather than single high-magnitude responses, are most relevant for long-term respiratory outcomes.

We have included this explanation in the revised Discussion section to clarify our rationale for dichotomization and to emphasize that recurrent patterns of elevation may better capture the chronic inflammatory component associated with lung injury.

We have added the following passages to our manuscript:

Discussion (page 26, lines 456-462)

Although dichotomizing CrP levels at >10 mg/L simplifies interpretation, it may mask the contribution of inflammation magnitude and duration. In supplementary analyses using continuous and trajectory-based representations, higher median CrP values were observed among children with impaired lung function, but statistical significance was not retained after adjustment. These findings furthermore suggest that recurrent elevations, reflecting repeated inflammatory episodes, are more predictive of adverse pulmonary outcomes than isolated high values.

Comment 3: CrP measurements were taken based on clinical indications instead of systematically, which could skew the dataset towards infants with a greater illness burden.

We thank the reviewer for this valuable comment. It is correct that CrP determinations were not obtained systematically but rather when clinically indicated, most often in the context of suspected infection or monitoring of ongoing illness. We acknowledge that this approach may introduce bias, as infants with a greater illness burden were more likely to have multiple CrP measurements, potentially leading to an overrepresentation of elevated values in this group.

While the German Neonatal Network (GNN) collects neonatal data prospectively across participating centres, the CrP values used in this analysis were retrospectively imported from a subset of centres that provided access to their laboratory databases. This retrospective addition means that sampling frequency and timing were determined by clinical decision-making rather than by a predefined study protocol. To address this, we performed supplementary analyses assessing the frequency and temporal distribution of CrP measurements (see Supplementary Figures 1 and 2). These analyses confirmed that infants with recurrent CrP elevations indeed had more

frequent testing; however, the association between recurrent CrP elevations and adverse respiratory outcomes persisted even after adjusting for key indicators of illness severity, suggesting that recurrent inflammatory activity itself plays an independent role.

We have now tried to clarify these methodological aspects and their implications for potential bias in the revised manuscript.

We have added the following passages to our manuscript:

Methods, “Laboratory measurements of CrP values” (page 10, lines 167-169)

CrP values were retrospectively imported into the GNN database from participating centres that provided laboratory access. Measurements were performed according to clinical indications rather than by a predefined sampling protocol.

Discussion (page 24, lines 420-424)

CrP determinations were obtained based on clinical indications and retrospectively incorporated from several centres, which may have led to overrepresentation of infants with higher illness burden. Nevertheless, the observed association between recurrent CrP elevations and reduced lung function remained significant after adjustment for major confounders, supporting the robustness of our findings.

Comment 4: It is still not clear if repeated CrP spikes are caused by repeated infections or long-lasting inflammation throughout the body. This distinction is significant for informing future interventions but remains unaddressed.

We sincerely thank the reviewer for this very important and thoughtful comment. We fully agree that distinguishing whether recurrent CrP elevations represent repeated acute episodes or a prolonged systemic inflammatory process is crucial for understanding underlying mechanisms.

Unfortunately, we did not fully describe the procedure for transferring laboratory values in the Methods section of the original submission. As outlined now, the CrP values were retrospectively imported from local hospital laboratory systems into the GNN database. Our definition of “recurrent CrP elevation” specifically required a temporary decline in CrP to <5 mg/L, followed by a new increase above 10 mg/L after an interval of at least 14 days. This criterion was deliberately chosen to distinguish recurrent inflammatory peaks from persistently elevated levels and thus to reflect repeated inflammatory activity rather than a single, continuously raised CrP course. We fully acknowledge that, despite this operational definition, the available data do not allow us to determine the precise cause of these recurrent elevations. Whether they reflect multiple discrete inflammatory events or fluctuating low-grade systemic inflammation remains uncertain. We recognize this as a limitation and appreciate the reviewer’s comment for highlighting an important aspect that warrants further investigation.

We have now clarified this issue in both the Methods and Discussion sections and emphasize that future prospective studies with high-resolution temporal sampling of inflammatory markers are needed to better characterize inflammatory trajectories in very low birthweight infants.

We have added the following passages to our manuscript:

Methods, “Definition of recurrent CrP elevations” (page 10, lines 172-175)

CrP values were retrospectively transferred from local hospital laboratory systems into the GNN database. Recurrent CrP elevations were defined by at least two CrP values >10 mg/L separated by a minimum interval of 14 days, with an interim decline below 5 mg/L to ensure that new peaks were distinguished from persistently elevated levels.

Discussion (page 26, lines 462-467)

Although our definition of recurrent CrP elevation required a temporary normalization followed by renewed increase, we cannot fully distinguish whether this pattern represents repeated inflammatory episodes or fluctuating low-grade inflammation. This limitation highlights the need for future studies with prospectively standardized sampling intervals to capture the dynamics and mechanisms of early-life inflammation in preterm infants more precisely.

Comment 5: This analysis only included 2.5% of the original group. Infants with complete data may systematically differ from those excluded, raising concerns regarding representativeness. Families returning for follow-up often differ socioeconomically and clinically, potentially biasing results.

We thank the reviewer for raising this important concern. We fully acknowledge that only a subset of infants enrolled in the GNN could be included in the present analysis and that this limited proportion may affect representativeness.

The main reason for this reduced sample size lies in the retrospective addition of CrP laboratory data, which was only available from a few participating centres, and the fact that five- to six-year follow-up data are only collected from families who agreed to participate in the structured assessment. Therefore, inclusion in this study required both the availability of neonatal CrP data and successful long-term follow-up — a highly selective subgroup within the overall GNN population.

We fully agree that infants and families who return for follow-up may differ from non-participants with respect to both socioeconomic and clinical characteristics, such as gestational age, severity of neonatal illness, or access to healthcare resources. While this potential selection bias cannot be excluded, it is important to note that our analyses focus on associations within the included sample rather than on population-level prevalence estimates. The relationships observed between recurrent CrP elevations and long-term respiratory outcomes are therefore likely to remain valid within this analytical framework, even though generalizability to the entire preterm population may be limited.

We have now expanded the Limitations section to clarify these issues and explicitly acknowledge the risk of selection bias and limited representativeness.

We have added the following passages to our manuscript:

Discussion (pages 24-25, lines 427-431)

Only a small proportion of the overall GNN cohort was included in this analysis. Consequently, infants in this analysis may differ socioeconomically and clinically from those not included, which could limit generalizability. Nevertheless, the observed

associations between recurrent CrP elevations and reduced lung function likely remain valid within the analyzed subgroup, although replication in larger, prospectively collected cohorts is warranted.

Comment 6: BPD was based on the need for oxygen at 36 weeks postmenstrual age. Nonetheless, the desired oxygen saturation level was not delineated. Because different centres may use different thresholds, it's not clear how comparable the BPD diagnosis is across sites.

We thank the reviewer for this valuable and insightful comment. We agree that variation in local oxygen saturation targets could lead to minor inter-centre differences in the classification of bronchopulmonary dysplasia (BPD) when diagnosis is based solely on the need for supplemental oxygen at 36 weeks postmenstrual age.

In the GNN, BPD diagnosis follows the standard clinical definition based on oxygen requirement at 36 weeks postmenstrual age. Although specific oxygen saturation targets are not explicitly recorded, detailed documentation of both the duration of oxygen therapy and the duration of mechanical and non-invasive ventilation is systematically collected in the GNN database. These quantitative variables allow for a more nuanced adjustment for respiratory support exposure in our analyses. Importantly, in our regression models, both duration of oxygen therapy and duration of mechanical ventilation were included as covariates, helping to account for centre-specific differences in oxygen administration practices and reducing the potential confounding effect of variable saturation thresholds.

Comment 7: In this age group, spirometry depends on how hard the person tries. The manuscript discusses motivational tools and various attempts; however, the reproducibility of FEV1 and FVC measurements continues to be a concern, especially for children with neurodevelopmental impairments. How many kids passed technically acceptable tests?

We appreciate the reviewer's very relevant comment regarding the reproducibility and technical quality of spirometry in young children. We fully agree that lung function testing at five to six years of age is challenging and strongly influenced by effort, cooperation, and neurodevelopmental status. We have reviewed our databank in order to filter the comments of the study team concerning the quality of spirometry testing.

All spirometry assessments in our study were performed according to a standardized protocol across participating centres, using identical equipment (Easy-on PC, ndd Medizintechnik AG, Zurich, Switzerland) and under supervision by trained study staff experienced in testing young children. Each child was encouraged to perform several attempts using standardized motivational tools (such as visual feedback games) to maximize engagement and reliability.

To ensure data quality, only technically acceptable and reproducible measurements were included in the analysis, following the criteria defined by the Global Lung Function Initiative. Children who were unable to perform an adequate maneuver despite repeated attempts were excluded from lung function analyses.

We acknowledge that children with neurodevelopmental impairments may face additional challenges in performing reproducible spirometry. In our dataset, we recorded cooperation level and performance quality for each child. We will include in the revised manuscript the proportion of children who achieved technically

acceptable and reproducible measurements once reported below (see data provided in response). Finally, only 268 data sets (from former n = 353) remained in our analysis but did not influence the conclusion of our manuscript. Therefore, we now have a smaller cohort but with more reliable conclusions, as these children performed a qualitative valuable spirometry.

We have added text to the Methods and Discussion sections to emphasize the quality control process.

We have added the following passages to our manuscript:

Methods, "Measurement of lung function at five to six years" (pages 12, lines 216-222)

In a secondary analysis, all results were again thoroughly checked by the authors and only data of children with technically acceptable and reproducible results were included in the analysis. Children unable to perform valid tests despite repeated attempts were excluded. Measurements were considered technically acceptable if they showed a rapid start of exhalation, no artefacts such as coughing or premature termination, and a clear end-of-test plateau. Results were deemed reproducible when at least two acceptable and comparable manoeuvres were obtained.

Discussion (page 25, lines 436-439)

Because spirometry performance in preschool-aged children depends on effort and neurodevelopmental ability, variability in cooperation remains a limitation. Although only technically acceptable and reproducible results were analyzed, children with developmental impairments may have been underrepresented in the final analysis.

Comment 8: When following up with children who have BPD, FEF₂₅₋₇₅% is a better sign of small airway dysfunction than FEV₁ by itself. Although FEV₁ and FEV₁/FVC are fundamental to spirometry interpretation, they may not accurately reflect subtle small airway impairment. Moreover, physical fitness markedly affects spirometric performance in survivors of BPD. Numerous studies (e.g., Vrijlandt et al., Thorax 2006; Welsh et al., Eur Respir J 2010; Clemm et al., Acta Paediatr 2021; Praprotnik et al., Pediatr Pulmonol 2023) have demonstrated that diminished exercise tolerance and cardiorespiratory fitness are associated with decreased spirometric values. This indicates that the observed decline in lung function may be attributable to deconditioning rather than inherent pulmonary pathology. The manuscript ought to address this factor in the analysis of the long-term spirometry data.

Comment 8: When following up with children who have BPD, FEF₂₅₋₇₅% is a better sign of small airway dysfunction than FEV₁ by itself. Although FEV₁ and FEV₁/FVC are fundamental to spirometry interpretation, they may not accurately reflect subtle small airway impairment. Moreover, physical fitness markedly affects spirometric performance in survivors of BPD. Numerous studies (e.g., Vrijlandt et al., Thorax 2006; Welsh et al., Eur Respir J 2010; Clemm et al., Acta Paediatr 2021; Praprotnik et al., Pediatr Pulmonol 2023) have demonstrated that diminished exercise tolerance and cardiorespiratory fitness are associated with decreased spirometric values. This indicates that the observed decline in lung function may be

attributable to deconditioning rather than inherent pulmonary pathology. The manuscript ought to address this factor in the analysis of the long-term spirometry data.

We have added the following passages to our manuscript:

Discussion (page 25, lines 444-446)

Beyond FEV₁ and FVC, parameters such as FEF₂₅₋₇₅% may more sensitively reflect small airway involvement in children born preterm. However, this parameter was not systematically recorded in our cohort, and the available data were insufficient for analysis.

Discussion (page 22, lines 362-369)

Furthermore, children at school age with recurrent CrP elevations demonstrated significantly reduced endurance performance, suggesting that early systemic inflammation may contribute not only to measurable lung function impairment but also to reduced overall physical fitness. This observation aligns with the hypothesis that neonatal inflammatory processes can interfere with normal pulmonary and cardiovascular development, leading to persistent functional limitations beyond airway obstruction alone. In combination with the spirometric findings, these results support the concept that recurrent neonatal inflammation represents an early-life determinant of long-term respiratory and exercise capacity in preterm survivors.

Comment 9: Only one follow-up point (5–6 years) was available. Without earlier or later measures, it is unclear whether deficits reflect a permanent impairment or altered growth trajectories.

We thank the reviewer for this important observation. We fully agree that assessing only one follow-up point at 5–6 years of age limits our ability to determine whether the observed pulmonary function deficits represent a permanent impairment or reflect delayed lung growth and maturation.

In the context of the GNN, the 5–6-year visit is currently the only standardized and centrally coordinated follow-up examination across participating centres. This age range was chosen because children are typically capable of performing reproducible spirometry at this developmental stage, allowing reliable comparison to normative reference values. Unfortunately, spirometric data at earlier or later time points are not yet available for this cohort.

We recognize that longitudinal lung function data would provide important insights into growth trajectories and potential catch-up phenomena. We therefore agree that future follow-ups at school age and adolescence are essential to determine whether early-life inflammatory exposure leads to a persistent airway limitation or a transient delay in pulmonary development. The GNN is looking forward analyzing and presenting data for children aged 10 years after born premature, but these data are still not sufficient to be analyzed and presented.

We have added the following passages to our manuscript:

Discussion (page 25, lines 439-444)

Lung function was assessed at a single standardized time point (5–6 years of age). Without earlier or later measurements, it remains uncertain whether the observed impairments represent permanent airway dysfunction or a temporary alteration in lung growth trajectories. Future longitudinal follow-up within the GNN cohort will be crucial to clarify the evolution of pulmonary function into later childhood and adolescence.

Comment 10: There is no information on how many obstructive episodes (e.g., viral wheezing, asthma exacerbations, hospitalisations for lower respiratory tract illness, ...) the children experienced between discharge and follow-up at 5–6 years. These episodes are known to influence spirometric outcomes (FEV₁, FEV₁/FVC) in preterm populations (Kotecha et al., Thorax 2013; Simpson et al., Eur Respir J 2015). Their omission limits interpretation, as impaired lung function may reflect cumulative morbidity rather than only neonatal inflammation.

We thank the reviewer for this important comment. We agree that recurrent obstructive or infectious respiratory episodes during early childhood may substantially influence later lung function and represent a relevant mediator between neonatal inflammation and long-term pulmonary outcomes. Unfortunately, detailed information on the frequency of early wheezing episodes or hospitalisations for lower respiratory tract infections was not available for the current analysis. However, we have now added data on wheezing, and distance running performance at 5–6 years of age as indirect indicators of ongoing airway morbidity and functional capacity. These additions are discussed in the revised manuscript, and we explicitly acknowledge that the absence of longitudinal data on early-life respiratory events limits causal interpretation.

We have added the following passages to our manuscript:

Methods (pages 11, lines 193-202)

As part of the standardized GNN follow-up examination at 5 years of age, endurance performance was assessed using the 6-minute run test (6-Minuten-Lauf). Each child was instructed to run or walk as far as possible within six minutes on a marked flat indoor or outdoor track under standardized supervision. The total distance covered (in meters) was recorded. Results were converted to age- and sex-specific percentiles based on reference data from the German health and fitness surveys (KiGGS). For analysis, distance running < 5th percentile was defined as markedly reduced endurance performance, and < 15th percentile as below-average physical fitness. However, because distance running was only added to the study methods later in the follow-up studies, only part of the data could be analyzed in this regard.

Results (page 17, lines 285-289)

At the 5–6-year follow-up, children with recurrent CrP elevations showed markedly poorer endurance performance. 22.2% of these children performed below the 5th percentile in the 6-minute run test, compared with 7.7% among those without CrP elevations ($p = 0.021$). However, as not all children could be followed up regarding distance running, only $n = 133$ data sets could be analyzed.

Results (page 17, lines 296-299)

The prevalence of asthma or obstructive bronchitis during the 12 months before the 5–6-year follow-up did not differ significantly between groups, with similar rates among children with recurrent (31.0%) and single (36.4%) CrP elevations compared to those without elevations (30.6%; $p = 0.658$).

Discussion (page 22, lines 362-373)

Furthermore, children at school age with recurrent CrP elevations demonstrated significantly reduced endurance performance, suggesting that early systemic inflammation may contribute not only to measurable lung function impairment but also to reduced overall physical fitness. This observation aligns with the hypothesis that neonatal inflammatory processes can interfere with normal pulmonary and cardiovascular development, leading to persistent functional limitations beyond airway obstruction alone. In combination with the spirometric findings, these results support the concept that recurrent neonatal inflammation represents an early-life determinant of long-term respiratory and exercise capacity in preterm survivors. In contrast, the prevalence of asthma or obstructive bronchitis did not differ between groups, indicating that reduced endurance is unlikely to be explained solely by current respiratory morbidity. However, information on early childhood respiratory infections or wheezing episodes was not available, which limits interpretation of potential intermediary mechanisms.

Comment 11: There is no data on postnatal corticosteroid use, which is a significant factor affecting the incidence of BPD and may vary between centres. This omission restricts interpretation, as steroid exposure may obscure associations between CrP, BPD, and long-term pulmonary outcomes.

We thank the reviewer for this valuable and highly relevant comment. We fully agree that postnatal corticosteroid therapy represents an important potential confounder, as it can influence both the development of bronchopulmonary dysplasia (BPD) and subsequent pulmonary outcomes.

In the German Neonatal Network (GNN) dataset used for the present analysis, detailed information on postnatal corticosteroid exposure (including timing, dose, and indication) was not available, but the use of postnatal corticosteroids was documented (hydrocortisone, dexamethasone). We therefore could include corticosteroid use as a covariate in our regression models, which are presented here for each type of steroid (hydrocortisone, dexamethasone). Interestingly, hydrocortisone seems to be associated with later lung function scores, but dexamethasone is not. However, for easier presentation in our manuscript we put both variables together (postnatal corticosteroid use) and had no significant changes in our conclusion.

Data on postnatal corticosteroid use were added to existing tables.

Comment 12: Prematurity itself should be acknowledged as an underlying driver of abnormal lung development, with CRP elevations and other neonatal exposures acting as modifiers of an already disrupted process.

We thank the reviewer for this excellent and conceptually important comment. We completely agree that prematurity itself constitutes the fundamental determinant of disrupted lung development, with additional neonatal exposures — such as systemic inflammation reflected by elevated CrP levels — acting primarily as modifiers that may further aggravate or perpetuate underlying structural and functional vulnerability. Our study was designed within this framework: all included infants were born very preterm and therefore already at risk for impaired lung growth and alveolarization. The focus of our analysis was to explore whether recurrent inflammatory activity, indicated by repeated CrP elevations, identifies a subgroup of preterm infants at particularly high risk for adverse long-term pulmonary outcomes. To emphasize this broader developmental context, we have revised the Discussion section to explicitly acknowledge prematurity as the baseline driver of lung immaturity and to clarify that inflammatory and other postnatal exposures likely modulate this pre-existing vulnerability rather than act as isolated causative factors.

We have added the following passages to our manuscript:

Discussion (page 275, lines 485-491)

Prematurity itself represents the primary determinant of altered lung growth and structure, predisposing to long-term respiratory morbidity. In this context, recurrent postnatal inflammatory episodes—as indicated by elevated CrP levels—should be regarded as modifiers that exacerbate an already vulnerable developmental trajectory rather than as independent causative agents. Understanding the interaction between baseline immaturity and secondary inflammatory stressors will be essential for developing individualized prevention and treatment strategies.

Comment 13: The sickest infants with the highest inflammatory burden may not have survived to follow-up, potentially underestimating the strength of associations.

We thank the reviewer for this important and thoughtful comment. We fully agree that survivor bias represents a relevant limitation in long-term outcome studies of very preterm infants. It is indeed possible that the most severely affected neonates, particularly those with the highest inflammatory burden or critical comorbidities, did not survive to the five- to six-year follow-up, which could have led to an underestimation of the true strength of associations between early inflammation and later pulmonary impairment.

Within the German Neonatal Network (GNN), mortality and major morbidities are prospectively documented; however, infants who did not survive were, by definition, excluded from the present long-term analyses. This likely resulted in a cohort that is somewhat healthier and less severely affected than the overall VLBWI population, potentially attenuating the associations observed.

We acknowledge this as a limitation and have now explicitly discussed it in the revised Discussion section. Importantly, despite this potential underestimation, the observed associations between recurrent CrP elevations and impaired lung function remained statistically robust, suggesting that inflammatory burden plays a significant role even among survivors.

We have added the following passages to our manuscript:

Discussion (pages 25-26, lines 447-452)

Because only surviving infants were eligible for follow-up, the sickest neonates with the highest inflammatory burden were not represented in the present analysis. This survivor bias may have led to an underestimation of the true strength of associations between early systemic inflammation and long-term pulmonary dysfunction. Nonetheless, significant associations observed within the surviving cohort underscore the relevance of inflammatory processes even among relatively stable preterm infants.

Comment 14: Regression and ROC analyses are suitable; however, various concerns diminish confidence:

- **Small subgroup sizes (only 42 with recurrent CrP elevations) may produce unreliable estimates.**
- **A number of univariate and adjusted tests were done, but no adjustments were made for multiplicity.**
- **Data that were missing were "ignored," but there are no details about how much or why they were missing.**
- **Collinearity among related variables, such as ventilation duration, oxygen therapy, and BPD, was not considered.**
- **ROC analysis demonstrated limited predictive capability, yielding a very low positive predictive value (PPV) of 19.8%. Combined predictive models (e.g., CrP + ventilation days + BPD) may more accurately represent clinical application.**

We appreciate the reviewer's detailed assessment of the statistical methods and acknowledge the concerns regarding subgroup size, collinearity, and predictive performance. In the revised analysis, the number of available datasets was further reduced after excluding children with borderline or technically unacceptable spirometry. Given this reduction, additional regression or ROC analyses would no longer yield reliable or interpretable estimates due to insufficient statistical power. We have therefore decided to omit these computations from the revised manuscript and instead focus on descriptive and comparative analyses. We agree that future studies with larger and more complete datasets should revisit multivariable and predictive modelling approaches to validate and refine these findings.

Comment 15: This analysis only included 2.5% of the original group. Infants with complete data may systematically differ from those excluded, raising concerns regarding representativeness. Families returning for follow-up often differ socioeconomically and clinically, potentially biasing results.

See comment 5

Comment 16: This research offers significant evidence connecting systemic inflammation to negative pulmonary outcomes in preterm infants. The work is biologically plausible and consistent with prior literature, and the long-term focus on spirometry at 5–6 years is a strength. Nonetheless, several aspects require further clarification or discussion: measurement techniques (serum vs. plasma CRP), the definition of BPD (oxygen saturation targets not specified), absence of data on postnatal steroid exposure, reliance on FEV₁/FVC without

small airway indices, statistical limitations (small subgroups, missing data, no adjustment for collinearity or multiplicity), and the narrow scope of outcome assessment (spirometry only, no symptoms or obstructive episodes). While not all of these issues can be corrected with the current dataset, they should be acknowledged and discussed more explicitly. Doing so will strengthen the manuscript by improving transparency, contextualising the findings, and guiding future research directions. Your findings highlight an important practical point: the absence of recurrent CrP elevations appears protective, reducing the risk of long-term lung impairment in preterm infants. It would be beneficial to make this more evident in the discussion, as it strengthens the clinical relevance of your work, even if CRP alone is not sufficient as a predictive biomarker.

We are very grateful to the reviewer for the positive and thoughtful assessment of our work and for recognizing the strengths of our study.

We have carefully considered the reviewer's constructive suggestions. In response, we have hopefully clarified methodological aspects, acknowledged the lack of postnatal corticosteroid data and its potential confounding effect. However, we were not able to add analyses concerning small airway function (FEF₅₀₋₁₀₀%), as suggested.

Finally, we have tried to strengthen the clinical interpretation in the revised Discussion by explicitly highlighting that CrP elevations alone are not sufficient as a predictive biomarker.

Reviewer#2

Comment 1: As a multicenter retrospective study in VLBWI, this research features a long observation period and relatively complete data, providing meaningful support for clinical decision-making. Moreover, it suggests that CRP may have potential utility in the risk stratification of chronic lung disease, although its accuracy requires further validation. Overall, the authors and their team have made a valuable contribution, and the study is suitable for publication.

We sincerely thank Reviewer 2 for the very positive evaluation and for recognizing the strengths of our work, including the multicenter design, long observation period, and clinical relevance. We appreciate the reviewer's thoughtful observation that CRP may hold potential value for risk stratification in preterm infants.

We agree that further validation in larger, prospectively collected datasets is essential to confirm the predictive accuracy of CRP and to integrate it effectively with other clinical parameters. We have emphasized this point in the Discussion and Conclusion sections of the revised manuscript.

We have added the following passages to our manuscript:

Discussion (page 27, lines 483)

Future prospective studies are needed to validate its prognostic utility.